# The Relationship between the Frequency and Duration of Physical Activity and Depression in Older Adults with Multiple Chronic Diseases

**DOI:** 10.3390/jcm11216355

**Published:** 2022-10-27

**Authors:** Jae-Moo Lee, Edward J. Ryan

**Affiliations:** 1Department of Sport Science, Sungkyunkwan University, Suwon 16419, Korea; 2Department of Exercise Science, Chatham University, Pittsburgh, PA 15232, USA

**Keywords:** depression, physical activity, chronic disease, older adults, mental health

## Abstract

Research has demonstrated that older adults with multiple chronic diseases (CD) are particularly vulnerable to depression. Meeting current recommendations for physical activity (PA) may help ameliorate the impact of depression on this population. Nonetheless, the impact of frequency versus duration of PA on depression in older adults remains to be explored. Therefore, the purpose of the present study was to determine the combined effect of PA and multiple CD on depression and the combined effect of the frequency, duration, and multiple CD on depression in older adults. Methods: The present study utilized data from the 2017 and 2020 Living Profiles of Older People Surveys. Data from a total of 19,907 older adults (10,042 older adults from 2017 and 9865 older adults from 2020) were included in the present study. Depression was assessed using the Korean version of the Short Form of Geriatric Depression Scale (K-SGDS) and CD included cardiovascular disease, respiratory diseases, thyroid syndromes, orthopedic complications, and diabetes. Participants who participated in PA ≥ 150 min/week were categorized as the high PA group, and those who participated in PA < 150 min/week were categorized as the low PA group. Furthermore, the frequency of PA (FRE) was divided into high FRE (≥5 times/week) and low FRE (<5 times/week), and duration (DUR) was divided into DUR30 (≥30 min/bout) and DUR0 (<30 min/bout). Results: The high PA group exhibited a lower risk of depression relative to the low PA group (*p* < 0.001). Furthermore, the risk of depression was consistently lower at DUR30 than DUR0 regardless of FRE in all CD categories and this result was maintained after adjusting for age, gender, BMI, height, weight, income, education levels, smoking status, and cognitive function. Conclusion: These results interestingly demonstrated that it is important for older adults to participate in a longer duration of PA to impact and prevent depression symptoms regardless of FRE.

## 1. Introduction

Depression is a mental and mood disorder that can cause several public health concerns and influence the quality of life [1]. According to the World Health Organization (WHO), depression is one of the common mental disorders impacting approximately 280 million people worldwide and represents one of the leading causes of non-fatal health loss [2]. The global prevalence of depression in older adults is approximately 10~20% [3,4]. This prevalence of depression in older adults is projected to continuously increase leading to intensifying social burdens and medical costs [5]. Unfortunately, depression is often neglected in the elderly population as it is often combined with other physical, physiological, and psychological disorders of aging. As a result, approximately 50% of depressive older adults are not being diagnosed or treated [3]. Importantly, untreated depression can increase the risk of other chronic diseases (CD), exacerbate existing diseases, increase mortality, and cause serious consequences such as suicide [6,7].

The causes of depression are various, including genetics, major life changes, stress, physical disorders, and medical disorders [8]. In older adults, depression often co-occurs with CD, and the incidence of depression increases in older adults with CD [9,10,11]. A previous study by DeJean et al. [12] reported that depression is closely associated with CD and a study by Alves and Rodrigues et al. [13] found that depression increased as the number of CDs increased.

Physical activity (PA) has been considered a promising option for older adults to maintain their healthy aging. PA is one of the most effective strategies to improve physical and psychological health and reduce mortality in older adults [14]. PA helps prevent and improve depressive symptoms and its’ antidepressant effect is consistently reported in the literature [15]. In addition, a recent cross-sectional study found that active older adults demonstrated better symptoms of depression than sedentary older adults [16]. In addition, PA is emphasized for older adults with CD, and it provides multiple health benefits and reduces symptoms of CD [17].

Although PA is beneficial for depression and chronic disease in older adults, there are still many older adults who are not engaged in PA. According to Kruger et al. [18], about 74% of older Americans did not meet the recommended criteria for PA and about 46% of older Americans were sedentary. The US Department of Health and Human Services [19] recommends ≥150 min/per week for older adults. This recommendation, ≥150 min/week is the resultant of the frequency (≥5/week) x the duration (≥30/each bout). However, it is still not clearly understood which frequency and duration should be more focused on to reduce the risk of depression in older adults with CD who are sedentary or who participate in PA less than the recommendation guideline. Accordingly, the present study will explore the combined effect of PA and multiple CD on depression and the combined effect of the frequency, duration, and multiple CD on depression in older adults.

## 2. Methods

### 2.1. Study Population

The present study utilized data from the 2017 and 2020 Living Profiles of Older People Surveys. This interview-based survey was conducted by the Korean Ministry of Health and Welfare to identify older adults’ living conditions and characteristics and improve their quality of life. Based on the Korean population census, a survey was conducted with about 10,000 people extracted using the population stratification sampling technique, and the participants of this study were selected by integrating data from 2017 and 2020. The interviewer personally visited each household and conducted a one-on-one direct interview. MMSE and SGDS were measured by recording direct responses to questionnaires. To assess physical activity, questionnaires were used to determine whether or not to do physical activity regularly, the frequency of physical activity per week, and the duration of the physical activity session. For chronic diseases, the total number of chronic diseases diagnosed by doctors among many chronic diseases such as obesity, diabetes, cardiovascular disease, and lung disease was directly confirmed through self-report. Data were collected from older adults who were aged 65 or higher and data from older adults who did not have key information including depression, CD, and PA were excluded. As such, a total of 19,907 older adults (10,042 older adults from 2017 and 9865 older adults from 2020) were selected for the present study.

### 2.2. Key Measurements

#### 2.2.1. Depression

To assess depression in Korean older adults, the Korean version of the Short Form of Geriatric Depression Scale (K-SGDS), which was based on SGDS developed by Sheikh and Yesavage [20], was applied [21]. The Cronbach’s alpha of K-SGDS to confirm the reliability was 0.880; it was 0.875 in the present study. The K-SGDS consisted of 15 questions related to behavioral symptoms of depression and 8 points or higher were categorized as depression [22].

#### 2.2.2. Multiple Chronic Diseases

Chronic disease was defined as one of 34 CD diagnosed by medical doctors including cardiovascular disease, respiratory diseases, thyroid syndromes, orthopedic complications, and diabetes, which was diagnosed by medical doctors, and the symptoms persist for more than 3 months. For the present study, there were 3 subgroups based on the number of CDs: no chronic disease group (CD0), 1 chronic disease group (CD1), and 2 or more chronic disease groups (CD2).

#### 2.2.3. Physical Activity

Physical activity levels were categorized as high PA and low PA based on the guidelines put forth by the US Department of Health and Human Services (US DHHS) suggesting PA ≥ 150 min/week [19]. The older adults who participated in PA ≥ 150 min/week were categorized as the high PA group and those who <150 min/week participated in PA were categorized as the low PA group.

In addition, PA levels are calculated by frequency/week × duration/each bout, and the US DHHS recommended 150 min/week with ≥5 of frequency/week and 30 min/each bout for older adults. Based on this formula, frequency (FRE) is divided into high FRE (≥5 times/week) and low FRE (<5 times/week), and duration (DUR) was divided into DUR30 (≥30 min/bout) and DUR0 (<30 min/bout).

#### 2.2.4. Covariates

Covariates included gender, age, cognitive function, body mass index (BMI), annual income, education level, and smoking. Cognitive function was obtained via the Korean version of the Mini-Mental State Examination (MMSE-K) [23]. In addition, gender consisted of 2 categories (male vs. female), education level consisted of 4 categories (no education vs. elementary school level vs. high school level vs. college level), and smoking consisted of 2 categories (current smokers vs. non-current non-smokers).

### 2.3. Statistical Analysis

For the present study, continuous variables are stated as means ± standard deviations, and categorical variables were reported as percentages. As mentioned above, the CD was divided into 3 groups defined as the CD0 group, the CD1 group, and the CD2 group based on the number of CDs and one-way analysis of variance (ANOVA), and the Chi-square test was conducted to compare the mean differences in variables among groups. In addition, PA was divided into 2 groups defined as the high PA group and the low PA group based on 150 min/week of recommended PA level, and an independent sample T-test and the Chi-square test were conducted to compare the mean differences in variables between groups.

To observe the combined effect of CD and PA on depression, a logistic regression analysis was performed to estimate the risks (OR, odds ratio) of depression. In addition, the risks of depression were adjusted for confounding factors (age, gender, annual income, education, cognitive function, and smoking). The reference group was the CD0 with a high PA subgroup (OR = 1).

Subsequently, to observe the combined impacts of FRE and DUR of PA on depression when combined with CD, 4 subgroups (the high FRE with high DUR group, the high FRE with low DUR group, the low FRE with high DUR group, and the low FRE with low DUR group) were applied to logistic regression analysis. The reference group was the high FRE with a high DUR group (OR = 1). In addition, adjusting for confounding factors was conducted. All statistical analysis was performed via SPSS version 22 (IBM, Chicago, IL, USA) and the significance was set a priori at *p* < 0.05.

## 3. Results

### 3.1. Multiple Chronic Diseases

Table 1 presents the comparisons of variables following the number of chronic diseases. While depression appeared more severe and cognitive function was more impaired as the number of CDs increased (*p* < 0.001), PA time decreased as the number of CDs increased (*p* < 0.001). In addition, the height decreased with the increase in the number of CDs (*p* < 0.001), while BMI increased with the number of CDs (*p* < 0.001). Concerning smoking status, there was a smaller number of smokers as the number of CDs increased (*p* < 0.001). In addition, as the education level increased, the number of CDs was more likely to decrease (*p* < 0.001).

### 3.2. Physical Activity

Table 2 presents a comparison of variables following PA. For depression, the high PA group demonstrated a lower risk of depression relative to the low PA group (*p* < 0.001). Similarly, the high PA group showed a lower risk of cognitive impairment than the low PA group (*p* < 0.001). The height, weight, and BMI were higher in the high PA group than in the low PA group (*p* < 0.001). The proportion of smokers was approximately 1% lower in the high PA group than in the low PA group (*p* < 0.001) and overall education levels were higher in the high PA group relative to the low PA group (*p* < 0.001).

### 3.3. The Combined Effect of Physical Activity and Multiple Chronic Diseases on Depression

Table 3 indicates the combined effects of PA and chronic diseases on depression. Compared to the high PA+CD0 (Model 1), the high PA+CD1 showed a ~3.1-fold increased risk of depression (*p* < 0.001). The high PA+CD2 showed a ~6.4-fold increased risk of depression (*p* < 0.001). For the low PA+CD0, the risk of depression was increased by ~1.9-fold (*p* < 0.001) and it was increased by ~3.6-fold in the low PA+CD1 (*p* < 0.001). In addition, the risk of depression was increased by ~11.9-fold in low PA+CD2 (*p* < 0.001). More importantly, the low PA+CD0 and the low PA+CD2 appeared to be approximately twice the risk of depression than the high PA+CD0 and the high PA+CD2, respectively (*p* < 0.001).

After adjusting for age and gender (Model 2, Table 3), the risk of depression was slightly reduced in all subgroups compared with prior to the adjustment; however, the trend was not altered. Furthermore, this trend was consistent after adjusting for other covariates including BMI, height, weight, income, education levels, smoking status, and cognitive function (Model 3).

### 3.4. The Combined Effect of Frequency and Duration of Physical Activity and Multiple Chronic Diseases on Depression

Table 4 shows the combined effect of the frequency and duration of PA and multiple chronic diseases on depression. Based on the high FRE+DUR30 (Model 1), in the CD0 category, the high FRE+DUR0 did not show a significant difference in the risk of depression *(p* = 0.224), and the low FRE+DUR30 reduced risk of depression by ~60% (*p* < 0.05). However, the low FRE+DUR0 demonstrated a ~2.0-fold increased risk of depression (*p* < 0.001). In the CD1 category, the high FRE+DUR30, the high FRE+DUR0, the low FRE+DUR30, and the low FRE+DUR0 demonstrated ~2.8 fold, ~3.9 fold, ~2.1 fold, and ~3.4-fold increased risk of depression, respectively (*p* < 0.001). In addition, in the CD2 category, the high FRE+DUR30, the high FRE+DUR0, the low FRE+DUR30, and the low FRE+DUR0 showed ~5.5-fold, ~9.3-fold, ~6.2-fold and ~11.1-fold increased risk of depression, respectively (*p* < 0.001). The important finding is that the risk of depression was consistently lower at DUR30 than at DUR0 regardless of FRE in all CD categories. This suggests that the duration of PA may be a more important factor to reduce the risk of depression in older adults than the frequency of PA. In addition, this result was maintained after adjusting for age and gender (Model 2) and adjusting for BMI, height, weight, income, education levels, smoking status, and cognitive function (Model 3).

## 4. Discussion

The present study investigated the relationships between FRE and DUR of PA and the risk of depression in older adults with multiple CD based on large-scale data from Living Profiles of Older People Surveys in Korea. In the present study, the risk of depression increased as the number of CDs increased, and while the total PA time increased, the risk of depression decreased. With respect to the impact of PA on depression in adults with CD, the results indicated that PA ameliorated the risk of depression at any given number of the CD. Furthermore, the data indicated DUR may be a more important factor relative to FRE in reducing the risk of depression in older adults with CD.

It has become clear that PA improves chronic health conditions, mobility, and mortality in older adults [24]. However, the antidepressant effect of PA in older adults has not been clearly understood. A systemic review by Mura and Carta [25] stated that there are inconsistent findings from previous studies and the knowledge of the effectiveness of exercise per se on depression in older adults has not been established. There may result from several confounding factors including different research protocols, study populations, and characteristics of depressive symptoms. Furthermore, the different doses of PA for depressive symptoms among previous observations might play a role in these contradictive results. For clinical applications of PA, a dose of PA consisting of optimal frequency, duration, modality, and intensity is a salient factor, and efforts to understand the details concerning depression and anxiety-related symptoms are warranted [26].

To extend our knowledge, we investigated the effects of frequency and duration of PA and chronic disease on depression and observed that duration seems to have a greater impact to reduce the risk of depression in older adults regardless of the number of chronic diseases. We view this finding as important in terms of PA in older adults. Based on a previous report, only about 26% of older Americans meet the recommended criteria for PA and about 46% of older Americans were sedentary [18]. Globally, 60% of older adults are sedentary and the proportion of sedentary behavior increases with aging [27]. Furthermore, according to the data in the present study, about 59% of older Koreans did not meet the recommended level of PA and about 40% of older Korean had a sedentary lifestyle. As a result, a significant proportion of older adults are still not engaged in PA or exercise inadequately. Therefore, the result of this study is meaningful in that it can be considered to maximize the beneficial effects of PA on depression for relatively inactive older adults. However, although this study found that the duration of physical activity is more likely to have a greater impact on depression in older adults than the frequency of physical activity, the intensity of physical activity was not included in the analysis. However, due to age-related limitations such as physical frailty, visual impairment, and cognitive deficit, it is somewhat difficult to engage older adults in vigorous-intensity of PA. Therefore, it is speculated that most older adults in this study had moderate-intensity PA. However, further study is needed to explore how the results of this study will be changed following the intensity.

The present study observed that PA is associated with a reduction in the risk of depression regardless of the number of chronic diseases. Supportive studies are reporting the effectiveness of PA on depression in patients with breast cancer [28] and patients with diabetes [29]. In addition, a review by Hare [30] stated that exercise was effective for depression with or without medications. Given these consistent results, it is speculated that PA can be a non-pharmacological therapeutic tool to reduce the risk of depression in older adults with multiple chronic diseases.

The present study has limitations. As mentioned above, the present study did not include exercise modality and intensity due to data availability that is part of the dose of PA; therefore, the results can be influenced by those variables. In addition, the study findings regarding frequency and duration are limited to the effect of PA on depression, and further studies are needed to determine the effect of frequency and duration of PA on other chronic health conditions. Although social relationships are recognized as an important factor closely related to depression in the elderly [31,32], this study focused on individual internal factors. Therefore, it is necessary to examine the relationship between depression, physical activity, and chronic disease, including social relationships among the elderly. This study utilized a large-scale dataset of the Korean older population; thus, the results of this study may not be generalized to other ethnic groups and/or countries. The data regarding depression, PA duration, cognitive function, and the number of chronic diseases were obtained via self-report, and as such this might influence the results of this study.

## 5. Conclusions

In conclusion, physical activity has been proven to be highly beneficial as a good non-pharmacological antidepressant to reduce the risk of depression symptoms in older adults although they have multiple chronic conditions. Moreover, to prevent depressive symptoms, it is important to participate in a longer duration of physical activity compared with frequent short bouts of physical activity participation.

## Figures and Tables

**Table 1 jcm-11-06355-t001:** Comparisons of variables following the number of chronic diseases.

	Number of Chronic Diseases (CD)				
* n *	0 * n * = 2640	1 * n * = 4512	≥2 * n * = 12,755	* p * Value	Post Hoc
	Mean ± SD	Mean ± SD	Mean ± SD		
PA time (min)	170.82 ± 206.60	158.44 ± 203.08	143.01 ± 187.05	<0.001	a > b > c
Depression (score)	2.00 ± 2.56	2.78 ± 3.04	4.45 ± 4.03	<0.001	a < b < c
Age (years)	71.23 ± 5.81	72.86 ± 6.24	74.88 ± 6.33	<0.001	a < b < c
MMSE-K (score)	25.64 ± 4.68	25.23 ± 4.61	24.27 ± 4.61	<0.001	a > b > c
BMI (Kg·m^2^)	22.78 ± 2.97	23.28 ± 3.18	23.90 ± 3.41	<0.001	a < b < c
Weight (Kg)	61.08 ± 8.75	61.17 ± 8.84	59.59 ± 9.28	<0.001	a, b > c
Height (cm)	162.50 ± 8.49	161.62 ± 8.39	158.36 ± 8.61	<0.001	a > b > c
Income (KRW 10,000)	1666.83 ± 2151.12	1543.93 ± 2262.04	1151.65 ± 1614.27	<0.001	a > b > c
Gender, * n * (%)					
Male	1392 (52.7)	2154 (47.7)	4435 (34.8)	<0.001	
Female	1248 (47.3)	2358 (52.3)	8320 (65.2)		
Smoking, * n * (%)					
Smoker	402 (15.2)	549 (12.2)	1074 (8.4)	<0.001	
None	2238 (84.8)	3963 (87.8)	11,681 (91.6)		
Education, * n * (%)					
None	239 (9.1)	526 (11.7)	3027 (23.7)	<0.001	
Elementary	720 (10.6)	1401 (20.6)	4690 (36.8)		
Middle-High	1457 (55.2)	2253 (39.0)	4436 (34.8)		
College	224 (13.3)	332 (28.7)	602 (4.7)		

Note: PA (physical activity); MMSE-K (Korean version of the Mini-Mental State Examination; BMI (body mass index); SD (standard deviation).

**Table 2 jcm-11-06355-t002:** Comparison of variables following physical activity.

	Physical Activity (PA)		
* n *	High * n * = 8280	Low * n * = 11,627	* p * Value
	Mean ± SD	Mean ± SD	
Depression(score)	3.09 ± 3.39	4.21 ± 3.97	<0.001
Age (years)	73.29 ± 5.98	74.89 ± 6.84	<0.001
MMSE-K (score)	26.65 ± 3.95	23.97 ± 4.98	<0.001
BMI (Kg/m^2^)	23.60 ± 2.67	23.51 ± 2.94	<0.001
Weight (Kg)	61.12 ± 9.06	59.45 ± 9.38	<0.001
Height(cm)	160.4 ± 8.49	158.87 ± 8.80	<0.001
Income (KRW 10,000)	1431.11 ± 1919.31	1221.69 ± 1867.61	<0.001
Gender, * n * (%)			
Male	3814 (46.1)	4167 (35.8)	
Female	4466 (53.9)	7460 (64.2)	<0.001
Smoking, * n * (%)			
Smoker	798 (9.6)	1227 (10.6)	
None	7482 (90.4)	10,400 (89.4)	<0.001
Education, * n * (%)			
None	1158 (14.0)	2634 (22.7)	<0.001
Elementary	2604 (31.4)	4207 (36.2)	
Middle & High	3807 (46.0)	4339 (27.3)	
College	711 (8.6)	447 (3.8)	

Note: PA (physical activity); MMSE-K (Korean version of the Mini-Mental State Examination; BMI (body mass index); SD (standard deviation).

**Table 3 jcm-11-06355-t003:** Combined effects of physical activity and chronic diseases on depression.

	Exposures		Model 1	* p * Value	Model 2	* p * Value	Model 3	*p* Value
	PA	CD	OR (95%CI)		OR (95%CI)		OR (95%CI)	
Total	High	CD0	1		1		1	
			(reference)		(reference)		(reference)	
	High	CD1	3.154	<0.001	2.977	<0.001	3.041	<0.001
			(2.195–4.531)		(2.070–4.279)		(2.111–4.380)	
	High	CD2	6.427	<0.001	5.644	<0.001	5.512	<0.001
			(4.595–8.988)		(4.032–7.899)		(3.931–7.730)	
	Low	CD0	1.98	<0.001	1.908	<0.001	1.74	<0.001
			(1.331–2.946)		(1.281–2.841)		(1.166–2.597)	
	Low	CD1	3.652	<0.001	3.222	<0.001	2.997	<0.001
			(2.569–5.190)		(2.264–4.584)		(2.102–4.273)	
	Low	CD2	11.918	<0.001	9.439	<0.001	8.392	<0.001
			(8.560–16.594)		(6.766–13.166)		(6.004–11.729)	

Statistical models are as follows: Model 1: no adjustment; Model 2 adjusted for age and sex; Model 3: adjusted for BMI, height, weight income, education levels, smoking, and MMSE score. Note: PA (physical activity); CD0 (no chronic disease group); CD1 (1 chronic disease group); CD2 (2 or more chronic disease groups); OR (odds ratio).

**Table 4 jcm-11-06355-t004:** Combined effects of frequency and duration and the number of diseases on depression.

	Exposures		Model 1	* p * Value	Model 2	* p * Value	Model 3	*p* Value
	FRE DUR	CD	OR (95%CI)		OR (95%CI)		OR (95%CI)	
Total	High FRE	CD0	1		1		1	
	DUR30		(reference)		(reference)		(reference)	
	High FRE	CD0	1.944	0.224	1.784	0.291	1.461	0.493
	DUR0		(0.666–5.567)		(0.609–5.227)		(0.495–4.314)	
	Low FRE	CD0	0.444	<0.05	0.466	<0.05	0.508	0.076
	DUR30		(0.211–0.934)		(0.221–0.981)		(0.241–1.072)	
	Low FRE	CD0	1.986	<0.001	1.938	<0.01	1.774	<0.01
	DUR0		(1.302–3.031)		(1.269–2.959)		(1.159–2.716)	
	High FRE	CD1	2.792	<0.001	2.638	<0.001	2.705	<0.001
	DUR30		(1.891–4.122)		(1.786–3.897)		(1.828–4.005)	
	High FRE	CD1	3.917	<0.001	3.29	<0.001	3.073	<0.001
	DUR0		(2.027–7.568)		(1.698–6.376)		(1.578–5.984)	
	Low FRE	CD1	2.1	<0.001	2.077	<0.001	2.217	<0.001
	DUR30		(1.373–3.212)		(1.358–3.179)		(1.445–3.402)	
	Low FRE	CD1	3.39	<0.001	3.02	<0.001	2.834	<0.001
	DUR0		(2.328–4.935)		(2.072–4.401)		(1.940–4.139)	
	High FRE	CD2	5.521	<0.001	4.896	<0.001	4.791	<0.001
	DUR30		(3.863–7.891)		(3.423–7.003)		(3.342–6.868)	
	High FRE	CD2	9.333	<0.001	7.399	<0.001	6.762	<0.001
	DUR0		(6.304–13.818)		(4.987–10.978)		(4.545–10.062)	
	Low FRE	CD2	6.187	<0.001	5.521	<0.001	5.69	<0.001
	DUR30		(4.308–8.885)		(3.841–7.937)		(3.939–8.198)	
	Low FRE	CD2	11.083	<0.001	8.881	<0.001	7.912	<0.001
	DUR0		(7.794–15.759)		(6.235–12.651)		(5.543–11.295)	

Statistical models are as follows: Model 1: no adjustment; Model 2 adjusted for age and sex; Model 3: adjusted for BMI, height, weight income, education levels, smoking, MMSE-score. Note: FRE (frequency of physical activity); DUR (duration of physical activity); CD0 (no chronic disease group); CD1 (1 chronic disease group); CD2 (2 or more chronic disease groups); OR (odds ratio).

## Data Availability

The data sets analyzed during the current study are available from the corresponding author on reasonable request.

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
