# Peer review of "The Relationship between the Frequency and Duration of Physical Activity and Depression in Older Adults with Multiple Chronic Diseases"

_jcm, 2022, doi:10.3390/jcm11216355_

Round 1

Reviewer 1 Report

This paper is well-written, and the results are clearly presented. A shortcoming is treating physical activity only as minutes and frequencies, not considering its intensity or modes. It would be important that the authors give the reader more information on how physical activity (and other variables!) were measured, so that there is more information from which to draw conclusions.

Methods:

There needs to be a better description on the participants and on how the data were collected. Who were invited to participate and how were they chosen? How many were contacted, how many participated? How were the participants contacted and how were data collected, interviews or postal/ web-based questionnaires? Was everything self-reported?  How were MMSE and SGDS administered? What instrument was used to assess physical activity and what was considered as PA when asking about it? Was information on chronic diseases confirmed from records?

Why do you adjust the analyses (Model 3 with height, weight, and BMI? The information on height and weight is included in BMI.

Do you have any information on social relations? If they lived alone or with someone, for example. And particularly if there was any social component in PA participation.

Results:

When reporting the differences in background characteristics, it would be better and more precise to talk about differences between CD groups than say that when the number of diseases increased/ decreased. As this is a cross-sectional study, the use of the verb increase may be misleading.

Table 1: There is a percentage missing for Middle-High education among those with 1 CD.

The MMSE mean scores are rather low among participants in each CD group. How is the reliability of responses confirmed?

Discussion:

What can be the reason for the duration of PA being more important than the frequency?

Author Response

Responses to Editor and Reviewer Comments

We would like to thank the reviewer for his/her insightful comments and constructive criticism of the manuscript. We have addressed each of the specific concerns raised by the reviewers in the order that they were discussed in the comments. The requested changes have been made and designated in the red italicized text to facilitate the review. Line numbers are also provided for some responses (the line numbering restarts each page).

Reviewer Comments:
Reviewer #1:

  1. His paper is well-written, and the results are clearly presented. A shortcoming is treating physical activity only as minutes and frequencies, not considering its intensity or modes. It would be important that the authors give the reader more information on how physical activity (and other variables!) were measured so that there is more information from which to draw conclusions.

We strongly agree with your comment on the intensity and mode of physical activity did not examine in the present. Those two variables are very important in physical activity-related research but the original survey did not ask about the intensity and mode of physical activity because the survey was conducted mainly on the over quality of life in the elderly. It would be great if the survey included the question regarding the intensity and mode of physical activity and address the information as the limitation of the study.    

Page 2 lines 75-83 We added the information on how PA was measured.

This interview-based survey was conducted by the Korean Ministry of Health and Welfare to identify older adults’ living conditions and characteristics and improve their quality of life.”

  1. There needs to be a better description of the participants and on how the data were collected. Who were invited to participate and how were they chosen? How many were contacted, how many participated? How were the participants contacted and how were data collected, interviews or postal/ web-based questionnaires? Was everything self-reported?  How were MMSE and SGDS administered? What instrument was used to assess physical activity and what was considered as PA when asking about it? Was information on chronic diseases confirmed from records?

Thanks for the comment! We added the information regarding that information in the manuscript

Page 2 line 75

“Based on the Korean papulation census, a survey was conducted with about 10,000 people extracted using the population stratification sampling technique, and the participants of this study were selected by integrating data from 2017 and 2020. The interviewer personally visited each household and conducted a one-on-one direct interview. MMSE and SGDS were measured by recording direct responses to questionnaires. To assess physical activity, questionnaires were used to determine whether or not to do physical activity regularly, the frequency of physical activity per week, and the duration of the physical activity session. For chronic diseases, the total number of chronic diseases diagnosed by doctors among many chronic diseases such as obesity, diabetes, cardiovascular disease, and lung disease was directly confirmed through self-report.”

  1. Why do you adjust the analyses (Model 3 with height, weight, and BMI? The information on height and weight are included in BMI.

Thanks for your comments

Those three variables could be confounding variables that can separately affect the results which is why we adjusted for them    

  1. Do you have any information on social relations? If they lived alone or with someone, for example. And particularly if there was any social component in PA participation.

I strongly agree with your suggestions. Unfortunately, that information was not included in the content of this study. The social relations would be very interesting and important factors regarding participating PA and other health outcomes. Therefore, we are currently working on the manuscript regarding social relations and health outcomes. Therefore, we will add content through the limitations of the study.

  1. When reporting the differences in background characteristics, it would be better and more precise to talk about differences between CD groups than say that when the number of diseases increased/ decreased. As this is a cross-sectional study, the use of the verb increase may be misleading.

Thanks for your comments!

We do agree with your comments and the characteristics of CD Group would be important information but we main purpose of the study was to examine the number of CDs not what kind of CDs participants have and it was very difficult to report them in one single table. In addition, however, we do think that the expression explains using increased/decreased is very profer standard English in use and we have consulted with an English Native Speaker to check the flow and mean of the results section.

  1. Table 1: There is a percentage missing for Middle-High education among those with 1 CD.

Thanks for your catch. We added the information.

  1. The MMSE mean scores are rather low among participants in each CD group. How is the reliability of responses confirmed?

The validity of the MMSE questionnaire utilized in the study was validated in other studies [Ref. 23] but we did not include the reliability and validity information in the manuscript. Please read below the text.   

“the internal consistency obtained by Cronbach's coefficient alpha was 0.826. The inter-rater reliability and test-retest reliability were 0.968 (p<0.001) and 0.825 (p<0.001), respectively. It showed a significant correlation between the Clinical Dementia Rating (CDR) (r=-0.698, p<0.05) and the three full Korean versions of MMSE (r=0.839-0.938, p<0.001). The area under the receiver operator curve for dementia of the SMMSE-DS was larger than those of the three full Korean versions of MMSE (p<0.001). Age, education, and gender explained 19.4% of the total variance of SMMSE-DS scores.” 

  1. What can be the reason for the duration of PA being more important than the frequency?

Thanks for the comment,

To increase the benefits of participating in physical activity (i.e., more calorie burn, stress relief), biochemical and physiological responses to physical activity may be essential and it requires a certain duration of physical activity time. However, the point of our finding is more likely “It is important to have a certain amount of time when performing physical activity”.

Importantly, the 2018 US physical activity recommendation guideline indicated that any minutes of physical activity should be performed but interesting the findings of the study were in line with the 2008 PA recommendation guideline. It means that when individuals with CD should be followed the previous PA recommendation guideline instead of the current PA guideline to gain additional health benefits.

Reviewer 2 Report

The article presents the results of very interesting research, the purpose of which was to investigate the effect of the frequency and duration of physical activity on depression in older adults with multiple chronic disease.

However, I miss an analysis that takes into account the type of chronic disease. It is known that not all chronic diseases are accompanied by depression to the same degree, and the role of physical activity in reducing symptoms of depression does not have to be the same. The authors have so much material that such an analysis would be possible.

Author Response

Responses to Editor and Reviewer Comments

We would like to thank the reviewer for their insightful comments and constructive criticism of the manuscript. We have addressed each of the specific concerns raised by the reviewers in the order that they were discussed in the comments. The requested changes have been made and designated in the red italicized text to facilitate the review. Line numbers are also provided for some responses (the line numbering restarts each page).

Reviewer Comments:
Reviewer #2:

However, I miss an analysis that takes into account the type of chronic disease. It is known that not all chronic diseases are accompanied by depression to the same degree, and the role of physical activity in reducing symptoms of depression does not have to be the same. The authors have so much material that such an analysis would be possible.

Thank you very much for your comments and we strongly agree with your comments.

There are many chronic diseases, and depending on the severity of each chronic disease, there will certainly be differences in depression. Also, the effect of exercise may differ depending on the characteristics of each chronic disease. However, Alves & Rodrigues et al. showed that the increase in the number of chronic diseases is significantly related to depression. Moreover, in the case of the elderly 65 years of age or older in the United States, about 70% of the population has two or more chronic diseases, and the US Department of Health and Human Services is further emphasizing the value of multi-chronic disease research. Therefore, the present study was conducted to examine the main effect of exercise on depression with the increase in the number of chronic diseases. To the best of our knowledge, this study would be the first study to examine the association between depression including in the number of chronic diseases and we strongly believed that the findings from this study would be additional information to the literature. 
